# Prevalence and Motivators of Getting a COVID-19 Booster Vaccine in Canada: Results from the iCARE Study

**DOI:** 10.3390/vaccines11020291

**Published:** 2023-01-28

**Authors:** Camille Léger, Frédérique Deslauriers, Vincent Gosselin Boucher, Meghane Phillips, Simon L. Bacon, Kim L. Lavoie

**Affiliations:** 1Department of Psychology, University of Québec at Montreal (UQAM), Montréal, QC C3H 3P8, Canada; 2Montreal Behavioural Medicine Centre, Centre Intégré Universitaire de Santé et de Services Sociaux du Nord de l’Ile de Montréal (CIUSSS-NIM), Montréal, QC H4J 1C5, Canada; 3School of Kinesiology, University of British Columbia, Vancouver, BC V6T 1Z1, Canada; 4Department of Health, Kinesiology & Applied Physiology, Concordia University, Montréal, QC H4B 1R6, Canada

**Keywords:** vaccine, booster, motivators, COVID-19, cross-sectional survey

## Abstract

Studies have shown that the protection afforded by COVID-19 vaccines against hospitalization and death decreases slowly over time due to the emergence of new variants and waning immunity. Accordingly, booster doses remain critical to minimizing the health impacts of the pandemic. This study examined the prevalence rate, sociodemographic determinants, and motivators of getting a COVID-19 booster vaccine within the Canadian population. We recruited a representative sample of 3001 Canadians aged 18+ years as part of the iCARE study using an online polling form between 20 January and 2 February 2022. Participants self-reported their booster status and were dichotomized into two groups: those who did vs. did not receive at least one booster dose. A total of 67% of participants received a booster dose. Chi-square analyses revealed that older age (*p* < 0.001) and having a chronic disease diagnosis (*p* < 0.001) were associated with being more likely to get a booster. Boosted individuals reported motivators tied to efficacy and altruism, whereas non-boosted individuals reported motivators tied to efficacy and safety. Results suggest that messaging will require careful tailoring to address the identified behavioral motivators among non-boosted individuals who emphasize safety and efficacy of additional vaccine doses.

## 1. Introduction

The early stages of the COVID-19 pandemic focused on reducing infection through the implementation of preventive behaviours and development of therapies and vaccines, which became available at the end of 2020 [1,2]. Vaccination against COVID-19 is one of the most important strategies to reduce disease-related morbidity and mortality [2]. However, there is increasing evidence that the effectiveness of COVID-19 vaccines against infection and mild symptoms can wane over time [3,4,5]. In light of this, the World Health Organisation Strategic Advisory Group of Experts on Immunization (WHO-SAGE) has recommended booster doses for people 4–6 months after their primary series of vaccination is completed [3]. Therefore, booster doses allow individuals to restore their protection after their primary vaccination series [6].

The success of COVID-19 booster immunization programs is dependent on the population’s willingness to receive the booster dose. Vaccine hesitancy is a major barrier to vaccine uptake, which is important for reducing the risk of disease-related morbidity and mortality, and to protect the most vulnerable in our population. Vaccine hesitancy, defined as the refusal or delayed acceptance of vaccination, was reported as one of the top ten threats to global health by the WHO [7,8,9]. As of 6 November 2022, in Canada, 80.4% of the population have completed the primary series of COVID-19 vaccines in comparison to 50% of the population who have received a 1st booster dose and 18.7% who have received a 2nd booster dose [10].

Few studies have examined the prevalence, sociodemographic determinants, and motivators associated with getting the COVID-19 booster vaccine across the world. A cross-sectional study in Jordan reported that uptake of booster doses included perceived side effects of the COVID-19 vaccine, perceived severity of COVID-19, and being at high-risk of developing COVID-19 complications [11]. Another study conducted in the United States demonstrated that attitudes and subjective norms about individuals’ beliefs in the value and utility of the booster vaccine, and the views of others who are important to them, were important determinants of intentions to get boosted [12]. A final study conducted in older adults in Israel found that older age and higher levels of education were associated with greater willingness to receive the COVID-19 booster vaccine, along with confidence in the healthcare system and the government [13]. Taken together, these studies reveal a wide range of motivators for booster vaccination across different countries, suggesting that they may not be universal. Despite a relatively high uptake of primary vaccine doses in Canada, much fewer have received booster doses, despite their availability (free of charge), and government recommendations to get boosted. To explore this phenomenon, we examined the prevalence, sociodemographic determinants, and motivators of getting a COVID-19 booster vaccine in a large population sample of Canadians.

## 2. Materials and Methods

### 2.1. Study Design

This study represents a sub-analysis of the International COVID-19 Awareness and Responses Evaluation (iCARE) Study (https://www.icarestudy.com, accessed on 28 December 2022) lead by members of the Montreal Behavioural Medicine Centre (MBMC: https://mbmc-cmcm.ca/, accessed on 28 December 2022). This international, cross-sectional, multi-wave observational study aims to examine public awareness, attitudes, and responses to COVID-19 public health policies through a series of online surveys. The study protocol and detailed methods have been published elsewhere [14]. The iCARE study was approved by the Research Ethics Board of the Centre Intégré Universitaire de Santé et de Services Sociaux du Nord-de-l’île-de-Montréal (CIUSSS-NIM), REB#: 2020-2099/25-03-2020.

### 2.2. Study Participants and Recruitment

For this sub-study, we analyzed one round of the Canadian representative sample, which was collected between 20 January and 2 February 2022. In order to collect reliable data on the Canadian population, samples were recruited through the proprietary online panel of Léger Opinion © (LégerWeb.com). This panel includes over 400,000 Canadians, the majority of which (60%) were recruited within the past 10 years. Two thirds of the panel were recruited randomly by telephone, with the remainder recruited via publicity and social media. Using data from Statistics Canada, the data were weighted within each province according to the sex and age of the respondents to make their profiles representative of the current population within each Canadian province (excluding the three territories). The weight of each province was further adjusted to represent their actual weight within the 10 Canadian provinces.

### 2.3. iCARE Survey Questionnaire

The survey includes approximately 75 questions and takes between 15–20 min to complete. The survey included questions about sociodemographics, physical and mental health, prior COVID-19 infection, general health behaviours, perceptions and attitudes about local COVID-19 prevention policies, concerns about the virus and its impacts, and vaccine attitudes, intentions, motivations, and behaviours. A detailed description and copy of all surveys can be found at the following weblink (https://osf.io/nswcm/, accessed on 28 December 2022). For the present report, we analzyed the following variables: sociodemographics; COVID-19 booster status; and motivators for getting the booster or not. For more information on the list of measures available in the iCARE study, as well our data dictionaries, see https://osf.io/c5rux, accessed on 28 December 2022.

COVID-19 booster status was assessed using the question “Have you received a COVID-19 vaccine booster dose?” (Possible responses: Yes, No, and I don’t know/prefer not to answer). In our analyses, participants were dichotomized into two groups: those who did vs. did not receive at least one booster dose. Motivators for getting a booster vaccine were also assessed by presenting participants with a list of 17 potential motivators (informed from the existing literature) [15,16], using the question: “To what extent did [for boosted] or would [for un-boosted] the following influence your decision to get a booster dose of the COVID-19 vaccine?” Responses were rated on a 4-point scale with the following response options: To a Great Extent; Somewhat; Very Little; Not at All; or I don’t know/I prefer not to answer. We only consider the response options “To a great extent” for this study.

### 2.4. Data Analysis

All analyses were weighted using data from Statistics Canada to approximate some degree of representativeness. Descriptive statistics were calculated to describe the sample in terms of sociodemographic characteristics. Bivariate analyses (chi-square tests) were conducted to examine the differences in sociodemographic characteristics (weighted proportions) as a function of getting the booster or not. The frequencies of participants who reported being motivated “to a great extent” for each motivator item was calculated as a function of getting the booster or not. Responses of ‘I don’t know/I prefer not to answer’ were treated as missing values.

Multivariate logistic regression was conducted to assess the association between having received a booster dose (independent variable) and motivators (dependent variable) while adjusting for covariates (sex, age, education, income, having a chronic disease or mental disorder, being a parent, being a healthcare or essential worker, COVID-19 infection, and weighting). All statistical tests were two-tailed and were performed using SAS version 9.4, with *p* < 0.05 being considered as statistically significant. 

## 3. Results

### 3.1. Sample Description

The total sample included 3001 individuals. Just over half the sample were women (52%), nearly half the sample were aged 51 and over (47%), and there was a high proportion who had an education level of high school or lower (73%). Of the sample, 67% received at least one booster dose. A summary of participant characteristics for the whole sample can be found in Appendix A.

### 3.2. Sociodemographic Differences as a Function of Booster Vaccination Status

Sociodemographic characteristics presented as a function of booster vaccination status are presented in Table 1. Chi-square analyses revealed that boosted individuals, compared to non-boosted individuals, were significantly older (60.5% compared to 27.9% over 51 years old; *p* < 0.001) and more likely to report being diagnosed with one or more chronic diseases (51.0% compared to 36.5%; *p* < 0.001). On the other hand, non-boosted individuals were more likely to report having a depressive (21.1% compared to 15.9%; *p* = 0.001) or anxiety (26.8% compared to 20.3%; *p* = 0.001) disorder, and were more likely to report being a parent (28.3% compared to 15.9%; *p* < 0.001) compared to boosted individuals. Finally, more non-boosted individuals reported being previously infected with COVID-19 (29.1% compared to 15.2%; *p* < 0.001) compared to the boosted group.

### 3.3. Motivators for Getting the COVID-19 Booster Vaccine

People who received a booster dose reported that the following were the most important motivators for their decision: (1) having information that the booster is effective (65%); (2) wanting to do their part to achieve “herd immunity” (65%); and (3) knowing that getting the booster would help protect others around them (70%). In comparison, people who had not received a booster reported that having information that the booster is: (1) effective (45%); (2) safe and unlikely to have any major long-term (44%); and (3) short-term (42%) side effects would motivate them to get vaccinated. The frequencies of motivators “to a great extent” as a function of booster vaccination status can be found in Appendix A and Figure 1.

A summary of the odds ratios for each motivator can be found in Figure 2 and Appendix A. Individuals who received the booster were three times more likely to be motivated by achieving “herd immunity” (OR = 3.27; 95% CI 2.62–4.07) and “helping protect others around them” (OR = 3.11; 95% CI 2.50–3.87) than individuals who did not get the booster. Individuals who received the booster were also twice as likely to be motivated by “following the recommended schedule” (OR = 2.53; 95% CI 2.00–3.20) and “wanting to reduce their worries and anxiety” (OR = 2.08; 95% CI 1.63–2.65) compared to individuals who did not get the booster. Individuals who received the booster were also between 1.55 and 1.80 times more likely to be motivated by “knowing the vaccine was effective against new COVID-19/variants” (OR = 1.80; 95% CI 1.45–2.24), “having information that it is effective against COVID-19” (OR = 1.78; 95% CI 1.43–2.21), “the convenience of getting the booster” (OR = 1.66; 95% CI 1.33–2.08), “seeing more people getting the booster” (OR = 1.65; 95% CI 1.28–2.14), “getting a recommendation from their doctor” (OR = 1.59; 95% CI 1.22–2.07), “hearing that other people have positive attitudes” (OR = 1.55; 95% CI 1.20–2.00), and “allowing them to go to restaurants/bars” (OR = 1.55; 95% CI 1.23–1.94) compared to individuals who did not get the booster. Individuals who received the booster were also between 1.35 and 1.40 times more likely to be motivated by “having information that it is safe (short-term)” (OR = 1.40; 95%CI 1.13–1.73), “having clear/consistent information from their government” (OR = 1.40; 95% CI 1.13–1.74), and “having information that it is safe (long-term)” (OR = 1.35; 95% CI 1.09–1.67) compared to individuals who did not get the booster. Conversely, individuals who received the booster were 23% less likely to be motivated by “having a choice about which booster I get” (OR = 0.67; 95% CI 0.53–0.84) compared to individuals who did not get the booster. Motivators such as “trusting the company who developed the booster”, “believing that I am high risk”, and “getting a recommendation from my employer” (*p* > 0.1) were not significant motivators.

## 4. Discussion

This study assessed the prevalence, sociodemographic determinants, and motivators of getting a COVID-19 booster vaccine in a large population sample of Canadians. Overall, 67% of our sample received a booster dose, which is slightly higher than previous reports [10]. Being older was associated with greater willingness to receive the booster. This is consistent with the published literature on COVID-19 and previous pandemics [11] and is likely related to the fact that older people have a substantially increased risk of morbidity and mortality from COVID-19 and have been prioritized to receive booster doses [17]. Chronic disease patients are also at a higher risk and thus prioritized, which may also explain why they were more represented in the boosted group [17,18].

The COVID-19 pandemic has introduced an unprecedented level of stress related to uncertainty and safety concerns. Intolerance to uncertainty can increase anticipatory anxiety regarding a perceived threat [19,20], and people who have intolerance towards uncertainty also often have difficulties making decisions [20,21]. Non-boosted individuals reported having more depressive and anxiety disorders, which are characterized by higher levels of anxiety, worry, and difficulties making decisions. Since there were higher levels of depressive and anxiety disorders in the non-boosted group, this may have been associated with more safety concerns and reluctance to get a booster dose [20].

We also found that non-boosted individuals were more likely to be parents than boosted individuals. Early information about COVID-19 being a mild illness in children, low confidence in the vaccine, and concerns about the risks of side effects may have contributed to hesitancy among parents [22,23,24].

Non-boosted individuals were also more likely to report not having been previously infected with COVID-19, which may have reduced risk perceptions and the motivation to get boosted in this group. This is consistent with the Health Belief model which predicts that adopting certain health behaviours (such as vaccine uptake) depends on a person’s belief in a personal threat of an illness or disease [25].

Both boosted and non-boosted individuals reported that the effectiveness of the booster was one of the most important motivators, which is consistent with previous reports [12,13,26]. Individuals who received the booster were also three times more likely to report being motivated by wanting to help achieve “herd immunity” or help protect them, relative to individuals who did not get the booster. This suggests that a willingness to get additional COVID-19 vaccine doses may be related to recognizing the benefits of widespread vaccination for the larger community and having stronger altruistic values, which has implications for tailoring vaccine/booster campaign messaging. Unpublished data from the iCARE study revealed that Colombia reported having mostly altruistic motivators to get the booster vaccine [27], which may be due to their vaccine campaign strategy which focused on getting the vaccine to protect others and achieve herd immunity [28].

Boosters being safe and unlikely to have any major long-term or short-term side effects was reported by non-boosted individuals as their primary motivators for getting the booster, which is consistent with the previous literature [11]. Understanding a population’s attitudes, intentions, and motivators for getting a booster vaccine could help to prioritize communication targets and tailor the message format and content to address the identified behavioral motivators considering the possibility of a COVID-19 vaccination on an annual basis.

The results of the present study should be interpreted in light of some limitations, including selection, response, and social desirability bias. However, this study also has some strengths. We collected data during a crucial period of vaccination: after broad public health campaigns and the arrival of vaccine passports, and the dramatic rise in COVID-19 cases due to the Omicron variant during the Christmas holidays. Additionally, we explored a range of motivators that point to several potential intervention targets for interventions. 

## 5. Conclusions

The results indicate that people who are younger and healthier are less likely to get COVID-19 booster doses. Boosted individuals reported motivators tied to efficacy and altruism, whereas non-boosted individuals reported motivators tied to efficacy and safety (lack of side effects). Given that newer variants have significantly reduced the altruistic benefits of vaccination, results suggest that messaging will require careful tailoring to address the identified behavioral motivators, especially in those who have not accepted a booster dose.

## Figures and Tables

**Figure 1 vaccines-11-00291-f001:**
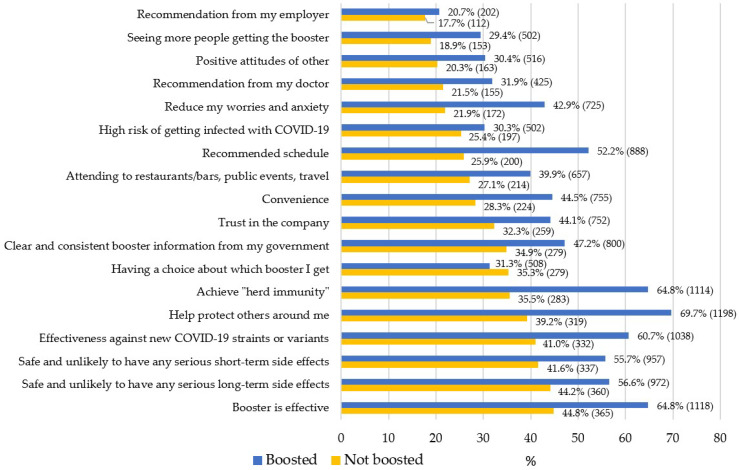
Frequencies of motivators “to a great extent” as a function of booster vaccination status.

**Figure 2 vaccines-11-00291-f002:**
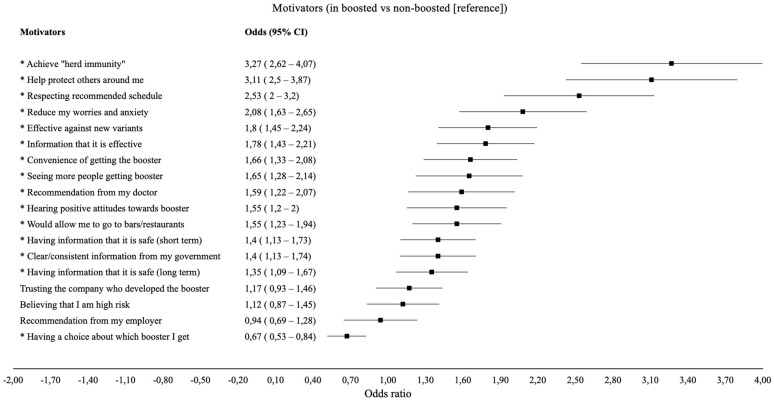
Adjusted * odds ratio for reporting each motivator “to a great extent” as a function of booster vaccination status in boosted vs non-boosted (reference), * = sign.

**Table 1 vaccines-11-00291-t001:** Participants’ characteristics as a function of booster vaccination status ^b^.

	Not Boosted	Boosted	
	(N = 859)	(N = 1744)	
**Descriptive characteristics variables**	**% (n)**	**% (n)**	** *p* ^a^ **
**Sex**			
Man	48.9 (419)	46.9 (813)	0.323
Woman	51.1 (437)	53.1 (921)	
Missing values	410		
**Age**			
Less than or equal to 25 years	18.7 (159)	9.5 (164)	**<0.001**
26–50 years	53.4 (455)	30.0 (522)	
51 years or more	27.9 (238)	60.5 (1051)	
Missing values	412		
**Education level**			
High school diploma or less	74.0 (633)	71.1 (1231)	0.132
College or more	26.0 (223)	28.9 (500)	
Missing values	415		
**Income**			
Less than 60K/year	47.6 (365)	43.7 (682)	0.075
60K/year or more	52.4 (401)	56.3 (877)	
Missing values	676		
**Chronic disease**			
No chronic disease	63.5 (529)	49.0 (836)	**<0.001**
At least one chronic disease	36.5 (304)	51.0 (870)	
Missing values	462		
**Presence of any depressive disorder**			
Yes	21.1 (175)	15.9 (273)	**0.001**
No	78.9 (651)	84.1 (1448)	
Missing values	454		
**Presence of any anxiety disorder**			
Yes	26.8 (223)	20.3 (347)	**0.001**
No	73.2 (610)	79.7 (1363)	
Missing values	458		
**Parent**			
Not a parent	71.7 (601)	84.1 (1444)	**<0.001**
Parent	28.3 (238)	15.9 (272)	
Missing values	447		
**Known or thought to have been infected with COVID-19**			
Yes	70.9 (227)	84.8 (248)	**<0.001**
No	29.1 (552)	15.2 (1383)	
Missing values	591		

^a^*p*-value from chi-square analyses; ^b^ Period: 20 January and 2 February 2022.

## Data Availability

The iCARE study data are available on request via the process identified here: https://mbmc-cmcm.ca/covid19/apl/ (accessed on 28 November 2022).

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
