# Peer review of "Prevalence and Motivators of Getting a COVID-19 Booster Vaccine in Canada: Results from the iCARE Study"

_vaccines, 2023, doi:10.3390/vaccines11020291_

Round 1

Reviewer 1 Report

An interesting publication allows for more intensive encouragement of social groups shown in the  work as less interested in continuing vaccinations

Author Response

We thank the reviewer for their very positive comments!

Reviewer 2 Report

Thank you for asking me to review this brief report.

Investigating the determinants of risk perception is crucial in the current pandemic context. The public perception of a health risk, in fact, plays a crucial role in influencing people's behavior and is often subject to the influence of social, cultural and psychological determinants, which, overall, constitute the main preconditions for its acceptability. Investigating the influence that the perception of a health risk has on the adoption of the correct preventive measures is a topic of considerable importance both for health surveillance and for the implementation of the best prevention strategies accompanied by adequate communication campaigns.

The study proposed by the authors is a Brief Report which intends to describe a series of secondary analyzes to a previous international study whose ultimate aim was to evaluate public awareness, attitudes and responses to public health measures adopted in order to combat the COVID-19 pandemic (such as administering vaccine and booster doses) by administering online surveys.

Overall, the study is well presented, however I believe that important elements have been overlooked which in the academic literature are described as important in the acceptance of the vaccine or its booster doses, first of all the phenomenon of Vaccine Hesitancy. Indeed, in the pandemic context, the phenomenon of vaccine hesitancy is further described above all in relation to its influence on health decisions (e.g. acceptability of administering a drug). I suggest to the authors to deepen this aspect in the introduction and to comment on it in the discussion section also in relation to the results obtained from their survey (doi: 10.3390/ijerph19074359).

Furthermore, being a study that describes secondary analyses, there are many references to the previous study within the text. This choice is understandable given the style of the manuscript, but, in my opinion, a large number of references to other previously published papers risks negatively affecting the originality of the Brief report. Perhaps the authors could think of deepening, for example, the macro-sections that describe the questionnaire administered by the authors in order to offer a general view without necessarily forcing the reader to deepen other manuscripts to get a clearer idea of the method.

I also suggest spending a few more words on how to recruit. How many participants were recruited by telephone? How many through new media? Was greater adherence found in the first or second recruitment modality? This could intrigue the reader also for the purpose of reproducibility of the study in other contexts.

The discussion is well presented and clearly commented, however the authors push a lot on the motivational aspects to get vaccinated rather than on the resistance to it. In my opinion, the results obtained must be placed in the more general context of vaccine hesitancy; in fact, this phenomenon needs renewed attention, especially in a historic moment such as the one the world population is experiencing given the status of COVID-19 pandemic (DOI: 10.1177/14034948221144661, doi: 10.3390/vaccines9111222).

Reviewer 3 Report

I'd like to thank for asking me to review this interesting paper which aim to examine the prevalence, sociodemographic correlates, and motivators of getting a COVID-19 booster vaccine in a large population sample of Canadians.

The authors should read these published papers on the same topic to deep discuss their findings:

DOI: 10.3390/vaccines10020141

doi: 10.3390/tropicalmed7120419

doi: 10.1038/s41467-022-28936-y

  •  

Author Response

  1. I'd like to thank for asking me to review this interesting paper which aim to examine the prevalence, sociodemographic correlates, and motivators of getting a COVID-19 booster vaccine in a large population sample of Canadians. The authors should read these published papers on the same topic to deep discuss their findings.

Response: We thank the reviewer for taking the time to read this brief report and for the suggestion to include these additional reports (which may have been published after we had submitted our manuscript). We have added the following references to the report: Trabucco Aurilio, 2022; Levine-Tiefenbrun, 2022; Coppeta, 2022. After reviewing all the references already cited and having added the references proposed by the different reviewers, we hope that we have improved the overall quality of the cited references relevant to the research as noted in the review report form.